# Proteomic Landscape of Extracellular Vesicles for Diffuse Large B-Cell Lymphoma Subtyping

**DOI:** 10.3390/ijms222011004

**Published:** 2021-10-12

**Authors:** Ana Sofia Carvalho, Henrique Baeta, Andreia F. A. Henriques, Mostafa Ejtehadifar, Erin M. Tranfield, Ana Laura Sousa, Ana Farinho, Bruno Costa Silva, José Cabeçadas, Paula Gameiro, Maria Gomes da Silva, Hans Christian Beck, Rune Matthiesen

**Affiliations:** 1Computational and Experimental Biology Group, CEDOC, Chronic Diseases Research Centre, NOVA Medical School, Faculdade de Ciências Médicas, Universidade NOVA de Lisboa, 1169-056 Lisbon, Portugal; henriquebaetah@gmail.com (H.B.); andreia.henriques@nms.unl.pt (A.F.A.H.); mostafa.ejtehadifar@nms.unl.pt (M.E.); 2Electron Microscopy Facility, Instituto Gulbenkian de Ciência, 2780-156 Oeiras, Portugal; etranfield@igc.gulbenkian.pt (E.M.T.); alsousa@igc.gulbenkian.pt (A.L.S.); 3iNOVA4Health—Advancing Precision Medicine, CEDOC—Chronic Diseases Research Centre, NOVA Medical School, Faculdade de Ciências Médicas, Universidade NOVA de Lisboa, 1169-056 Lisbon, Portugal; ana.farinho@nms.unl.pt; 4Systems Oncology Group, Champalimaud Research, Champalimaud Centre for the Unknown, 1400-038 Lisbon, Portugal; bruno.costadasilva@research.fchampalimaud.org; 5Pathology Department, Instituto Português de Oncologia de Lisboa Francisco Gentil, 1099-023 Lisbon, Portugal; jcabecadas@ipolisboa.min-saude.pt; 6Haematology Unit, Instituto Português de Oncologia de Lisboa Francisco Gentil, 1099-023 Lisbon, Portugal; pgameiro@ipolisboa.min-saude.pt (P.G.); mgsilva@ipolisboa.min-saude.pt (M.G.d.S.); 7Centre for Clinical Proteomics, Department of Clinical Biochemistry and Pharmacology, Odense University Hospital, DK-5000 Odense, Denmark; hans.christian.beck@rsyd.dk

**Keywords:** extracellular vesicles, exosomes, DLBCL, diffuse large B-cell lymphoma, proteomics, mass spectrometry

## Abstract

The role of extracellular vesicles (EVs) proteome in diffuse large B-cell lymphoma (DLBCL) pathology, subclassification, and patient screening is unexplored. We analyzed by state-of-the-art mass spectrometry the whole cell and secreted extracellular vesicles (EVs) proteomes of different molecular subtypes of DLBCL, germinal center B cell (GCB subtype), and activated B cell (ABC subtype). After quality control assessment, we compared whole-cell and secreted EVs proteomes of the two cell-of-origin (COO) categories, GCB and ABC subtypes, resulting in 288/1115 significantly differential expressed proteins from the whole-cell proteome and 228/608 proteins from EVs (adjust *p*-value < 0.05/*p*-value < 0.05). In our preclinical model system, we demonstrated that the EV proteome and the whole-cell proteome possess the capacity to separate cell lines into ABC and GCB subtypes. KEGG functional analysis and GO enrichment analysis for cellular component, molecular function, and biological process of differential expressed proteins (DEP) between ABC and GCB EVs showed a significant enrichment of pathways involved in immune response function. Other enriched functional categories for DEPs constitute cellular signaling and intracellular trafficking such as B-cell receptor (BCR), Fc_gamma R-mediated phagocytosis, ErbB signaling, and endocytosis. Our results suggest EVs can be explored as a tool for patient diagnosis, follow-up, and disease monitoring. Finally, this study proposes novel drug targets based on highly expressed proteins, for which antitumor drugs are available suggesting potential combinatorial therapies for aggressive forms of DLBCL. Data are available via ProteomeXchange with identifier PXD028267.

## 1. Introduction

B-cell lymphomas develop from B-lymphocytes and account for 85% of all non-Hodgkin lymphomas (NHLs). Diffuse large B-cell lymphoma (DLBCL) is the most common form of lymphoma, accounting for 25% to 30% of all newly diagnosed cases of NHL. DLBCL is also a heterogeneous entity, encompassing a number of morphologic variants, distinct biologic entities, and variable clinical behaviors and responses to treatment [1].

DLBCL is an aggressive form of NHL, infiltrating organs other than the lymph nodes at presentation in about 40% of the cases and with suboptimal outcomes in a fraction of patients. Additionally, some DLBCL subtypes are defined by organ location and have particular biological characteristics. The disease demonstrates remarkable progression-free survival (PFS) and overall survival (OS) rates in clinically and biologically defined subgroups of patients [2]. Unfortunately, approximately 40% relapse or develop refractory disease upon standard RCHOP (rituximab, cyclophosphamide, doxorubicin, vincristine, and prednisone) treatment. Gene expression profiling (GEP) is currently the gold standard for determining the cell of origin (COO). GEP of DLBCL revealed two main subtypes: GCB (germinal center B-cell like) and non-GCB. Non-GCB tumors include the unclassifiable and the activated B-cell like (ABC) DLBCL subtypes; the latter is associated with poor treatment outcomes [2,3]. In addition, a rarer primary mediastinal B-cell type was also identified by GEP.

The current limitations of the gold-standard method for DLBCL molecular subtyping (frozen tissue Affymetrix-based method) are associated with RNA isolation. Furthermore, its susceptibility to degradation by chemical treatments such as those used in formalin-fixed paraffin-embedded (FFPE) tissue preparation raises challenges for standardization. Fresh tissue biopsies are in some cases unavailable. Thus, the development of practical and reliable diagnostic methods derived from liquid biopsies has the potential to ameliorate the limitations of current methods and additionally improve monitoring of minimal residual disease. Current diagnostics based on immunohistochemistry (IHC), which serves as surrogate for COO, are not accurate despite the many algorithms proposed [4]; in addition, traditional gene expression profiling is not applicable to routine diagnosis. Newer approaches such as nanostring technology are not generally used due to cost and the need for good quality tissue RNA. In order for targeted therapies to succeed in DLBCL, minimal invasive robust strategies that segregate patients into molecular groups useful for treatment decision and with high reliability constitute an unmet clinical need [5].

Large-scale proteomics enabled clear separation of cell lines representing GCB and ABC subtypes [6,7]. Follow-up clinical studies on FFPE tumor tissues [8,9,10] have provided promising results. Nevertheless, the advantage of proteome profiling over gene expression profiling based on FFPE tumor tissue is still unclear. The potential of EV proteomes to discriminate between GCB and ABC remains to be investigated. The possibility to diagnose and screen DLBCL patients for treatment based on liquid biopsies by targeting the proteome of extracellular vesicles (EVs) is unexplored and provides clear advantages. Minimally invasive liquid biopsy-based diagnosis and patient screen opens new avenues for fast and longitudinal follow up for minimal residual disease assessment [11,12].

EVs have attracted much recent interest because of their potential functions, use as disease biomarkers, and possible therapeutic exploitation [12,13]. EVs are extracellular vesicles of endosomal and plasma membrane origin released in vivo into the extracellular environment by cells as distinct as B lymphocytes and dendritic cells as well as from several additional cell types of hematopoietic origin [14]. EVs have been shown to play fundamental roles in intercellular communication by carrying and selectively deliver bioactive molecules (e.g., lipids, proteins, and nucleic acids). EVs are tailor-made specialized mini-maps of their cell of origin and of both physiological and pathological relevance [11]. This feature leads to the potential to use EVs as blood-based biomarkers of diagnosis and prognosis for hematopoietic cancers.

We consequently questioned the potential of EV proteomes as a surrogate biomarker for DLBCL COO segregation. As a preclinical proof of concept, we applied large-scale mass spectrometry to profile the whole-cell and EV proteomes of four cell lines from GCB and ABC DLBCL subtypes.

## 2. Results

To explore the potential role of EVs proteome in the establishment of a signature to distinguish DLBCL subtypes based on COO, we have used established cell lines derived from patients annotated to DLBCL-GCB subtype (DB and HT) and ABC subtype (RIVA and OCI-Ly3). For each cell line, we have analyzed the whole-cell and derived EVs proteomes of three independent biological replicas. EVs were purified from DLBCL cells condition media using serial ultracentrifugation combined with floatation on sucrose cushion (UC-suc) [15]. The whole-cell and EVs lysates were analyzed by LC-MS in two technical replicates. Figure A1 summarizes the experimental outline. Results from multivariate analysis, functional enrichment, and data exploration are described in the subsequent sections.

### 2.1. Analysis of DLBCL Cell Lines Derived EVs

DLBCL EVs were characterized for size distribution, concentration, protein content, presence of contaminants, and morphology. By transmission electron-microscopy negative staining, we have observed vesicles with an artificial cup-shaped-like structure caused by fixation and dehydration during mounting preparation [14,16], suggesting that EVs preparations were considerably enriched in exosomes and microvesicles with an average size between 50 and 200 nm (Figure 1a). Size and concentration of EVs were also determined using nanoparticle tracking analysis and ranged from 80 to 240 nm (Figure 1b) and 0.4–1.0 × 10^9^ particles per microgram of protein and 1–3 × 10^11^ particles per milliliter (Table 1).

This corresponds to particle-to-protein ratio values concordant with the ratios reported by Webber and Clayton [17]. To further access quality of DLBCL EVs isolations, we compared our MS data with previously published EVs MS studies using our previous developed quality assessment tool (Figure 1c) [12]. We observed abundant expression of known EV markers comparable to previous large-scale MS-based EV characterization studies [18,19]. Additionally, lower-ranked protein expression levels of markers indicating contamination from other subcellular fractions such as the endoplasmic reticulum (CANX), golgi apparatus (GOLGA2), mitochondria (BCL2), and nucleus (NUP98) were in the same range as reported by others [18,19].

### 2.2. Overview of Identified Proteins

MS-based proteomics analysis of DLBCL whole-cell proteome and secreted extracellular vesicles identified in total 17,833 protein isoforms at a 1% FDR threshold. Collapsing the protein isoforms into their encoding genes resulted in 5949 proteins identified in all independent biological and technical replicates. Given the high level of replication (two technical replicas and three independent cell cultures), the statistical differences were accessed for the identified proteins across all cell lines. The total number of proteins within each cell line was highly reproducible considering both biological and technical replicates (Figure 2a). A significant difference was observed among identified proteins between GCB and ABC subtypes. OCI-ly3 displayed a significantly higher number of identified proteins compared to the other cell lines (significant difference only indicated for OCI-ly3 in comparison with DB, in Figure 2a). For all other comparisons, the significant differences in the number of identified proteins are indicated in Figure 2a. RIVA displayed lower total number of identified proteins than the GCB cell lines. Proteins identified uniquely in OCI-ly3 cell line (902 proteins) were enriched in the gene ontology cellular components: nucleolus (*p* value = 1.6 × 10^−12^), nucleoplasm (*p* value = 3.4 × 10^−7^), and mitochondrion (*p* value = 5.7 × 10^−7^). This result pinpoints high heterogeneity in DLBCL and is concordant with previously described heterogeneity based on genomic data [20]. It further highlights the potential of large-scale proteomics in further subclassifying DLBCL.

All DLBCL cell lines analyzed the number of identified proteins from EVs and were smaller than the corresponding identifications obtained from the cellular proteome (Figure 2b). Since EVs constitute subfractions of cells, it is expected that the EVs proteome content is a subpart of the cellular proteome. The number of identifications in EVs displayed larger variance than the cellular proteomes. This can also be a result of the larger number of steps required for isolation of EVs. Nevertheless, even for EVs the numbers of identified proteins were sufficiently reproducible to allow for the detection of significant differences between EVs obtained from different cell lines.

### 2.3. Functional Enrichment Analysis of Identified Proteins

Gene ontology cellular component based functional enrichment analysis of all identified proteins revealed clear clustering patterns of cellular compartment associated proteins specific for either EVs fractions or whole-cellular proteome (Figure 3). Cellular compartments such as extracellular vesicular exosomes, cytosolic, membrane, nucleoplasm, and nucleosome were highly enriched in both EVs and whole-cell proteomes. Proteins belonging to mitochondria-related compartments such as mitochondrion, mitochondrial matrix, and mitochondrial inner membrane were exclusively enriched in the whole-cell proteome. On the other hand, EV fractions were most enriched for proteins belonging to membrane-vesicle-related compartments such as phagocytic vesicle membrane, lysosomal membrane, recycling endosome membrane, and late endosome. In addition, while proteins of the proteasome complex were enriched in EVs, proteins of the regulatory particle 19S were less enriched in the vesicle fractions compared to the whole-cell proteome (Figure 3). An important aspect relevant for endosomal EV sorting of protein components are ubiquitin and ubiquitin-like modifiers; and therefore, the proteasome complex is of particular relevance. The identification of proteasome complex proteins in EVs has previously been described as extracellular [21]. However, further studies are required to assess luminal or outer membrane proteasome localization. Besides the nucleoplasm and nucleosome, all other nuclear subcellular components were mainly identified in the whole-cell proteome.

### 2.4. Multivariate Analysis of Quantitative MS Data

Statistical comparison between ABC and GCB quantitative proteomics data (Appendix A) revealed significance differences in 288/1115 expressed proteins (DEPs) from whole-cell proteome and 228/608 DEPs from EVs (adjust *p*-value < 0.05/*p*-value < 0.05). Volcano plots for the comparison between ABC and GCB whole-cell and EVs proteomes are provided in Figure 4a,b, respectively. At the cellular level, subsets of upregulated proteins as well as subsets of downregulated proteins were clearly separated from the general cloud of data points (Figure 4a). For EVs, mainly ABC upregulated proteins were clearly separated from the general cloud of data points in the volcano plot (Figure 4b). CD44, a cell-surface receptor involved in the response to changes in the tissue microenvironment, was the most significantly upregulated and displayed the highest log_2_ fold change in EVs (Figure 4b). Principal component analysis using all quantitative data without any bias filtering of quantitative values showed separation of DLBCL into COO subtypes for both whole cell (Figure A2a) and EVs (Figure A2b).

### 2.5. Functional Analysis of Differentially Expressed Proteins

KEGG functional analysis and GO enrichment analysis for cellular component, molecular function and biological process of DEPs between ABC and GCB EVs showed significant enrichment of pathways involved in immune response/function and concomitantly cellular signaling and intracellular trafficking such as B-cell receptor (BCR), Fc_gamma R-mediated phagocytosis, ErbB signaling, endocytosis, and natural killer cell mediated cytotoxicity (Figure 5). Proteins assigned to natural killer-cell-mediated cytotoxicity pathway overlap totally, except for ICAM1, with proteins in the remaining pathways. GO enrichment analysis for cellular component, the molecular function and biological process are summarized in Appendix A (whole cells) and Appendix A (EVs). We observed that EVs DEP proteins are mostly upregulated in the abovementioned pathways (Figure A3). After the comparison of DLBCL EV DEP in each pathway showed overlapping proteins between the five mostly enriched pathways such as proto-oncogene tyrosine-protein kinase SRC, E3 ubiquitin-protein ligase CBL, Casitas B-lineage lymphoma proto-oncogene b CBLB, and serine/threonine protein kinase PK3CA. On the other hand, there were proteins unique to each of the pathways such as the B-lymphocyte cell adhesion molecule, CD22, and the tyrosine-protein kinase BTK for B-cell receptor signalling. Furthermore, unique proteins associated to vesicle transport and cargo sorting such as multivesicular body proteins, ESCRT-I complex proteins, and RABs characteristic of the endocytic pathway were also observed. For EVs and for whole-cell proteome 21 and 15 proteins, respectively, were significantly regulated and associated to the B-cell receptor signaling pathway (Figure A3). Furthermore, the majority of EVs proteins associated to B-cell receptor signaling pathway were upregulated in ABC EVs. However, KEGG functions related to cancers, viral infection, proteasome, and focal adhesion were also enriched. The enriched KEGG pathways for EVs are to some extent interconnected with shared annotated proteins (Figure A4). For example, the ErbB signaling pathway had 18 shared regulated proteins with an insulin signaling pathway (Figure A4). Based on our data and literature evidence on activation of B-cell receptor, we can suggest that the endocytic pathway is upregulated in DLBCL ABC subtype compared to GCB. Furthermore, ABC cells may be preferentially active in sorting BCR-associated protein molecules, suggesting that the biology of this subtype has an important dependency on immunoregulatory pathways, influencing a response to treatment.

We have identified 288 DEPs between EV proteins of GCB and ABC cell lines (adj. *p* value ≤ 0.05). Among the top 20 most significantly expressed EV proteins are proteins annotated to the leukocyte transendothelial migration KEGG pathway. CD44 antigen, intercellular adhesion molecule 1 (ICAM1), and tyrosine-protein kinase JAK1 (JAK1) were upregulated in EVs of the ABC subtype. Furthermore, significantly regulated proteins (adjusted *p* value < 0.05) overlap with previously proposed biomarkers from NanoString technology and from the study by Deeb et al. [6] (Figure A5). In all cases, the direction in the regulation was concordant between both studies despite the use of two different quantitative MS methodologies (label free versus SILAC).

Overall, we identified a higher number of significantly regulated proteins than Deeb and coworkers [6]. This could be explained by differences in quantitative MS methodology, instrument sensitivity or the higher number of cell lines investigated in the study by Deeb et al. [6]. The overlap for the whole-cell proteome (Figure A5a) was higher than for EVs (Figure A5b) when compared to the whole-cell proteome from Deeb et al. [6]. The proteins reported in both studies stand out as highly reliable biomarker candidates.

NanoString technology is based on RNA expression in tumor tissue. Nevertheless, the 20 targets used by NanoString technology were compared to significantly differentially expressed proteins (adjusted *p* value < 0.05) from whole cells (Figure A5c) and EVs (Figure A5d), and a few concordant proteins were identified, namely IRF4, CYB5R2 and MME (whole cell), and CCDC50 (sEV).

Among the significantly regulated proteins (adjusted *p* value < 0.05), we identified a number of annotated tumor suppressors and oncogenes (Figure A6). Overall tumor suppressors were more abundant in GCB than ABC and oncogenes more abundant in ABC for whole-cell proteome data. The ABC EVs proteome displayed a high number of significantly upregulated proteins annotated as oncogenes.

### 2.6. Identification of Potential Proteostasis-Related Drug Targets

Deubiquitinases (DUBs) function as both pro-oncogenic factors and tumor suppressors in cancer and have been targeted for cancer therapy [22]. Therefore, the MS data were explored for potential drug targets. As an example, VLX1570 and an analog b-AP15 specifically block the activity of USP14 and UCHL5 in the 19S regulatory subunit and display robust antitumor activity in well-established orthotopic and xenograft models [23]. Both USP14 and UCHL5 were expressed in all four DLBCL cell lines in this study (Figure 6). Especially, USP14 was highly expressed in all cell lines. In total, 33 DUBs were identified of which eight were highly expressed in all cell lines. Of notice, OCI-ly3 showed the highest expression (higher log_2_ (ion counts) of DUB proteins comparatively with the remaining cell lines.

E3 ligases have also been proposed as specific drug targets given that there are more than 600 genes coding for E3 ligases in the human genome [24]. However, specific drugs targeting E3 ligases are still missing. The majority of the E3 ligases identified were scarcely expressed. Nevertheless, five were highly expressed in all cell lines (Figure A7). Similar to what was observed for DUB proteins, OCI-ly3 showed a broader spectrum of expression of different E3 ligases as well as higher log_2_(ion counts).

The protein expression values in cells were also matched against all approved drugs that targets specific proteins (Figure A8). The five proteins NPM1, CYB5R3, IDH1, HPRT1, and G6PD with approved drugs were observed abundantly expressed in all cell lines. All of these proteins are considered as promising targets for treatment of cancer and some specifically for hematological cancers [25,26,27,28,29].

## 3. Discussion

Gene expression profiling of DLBCL subclassifies the disease into at least two main molecular subtypes which correlate with patient outcomes [30]. The availability of high-quality RNA from tumor samples is cumbersome, and liquid biopsy-based methods represent a more practical approach in a clinical diagnostic setting. EVs carry molecular traits of the parental cells and therefore are a potential source of biomarker molecules for diagnosis and prognosis of DLBCL [13]. EVs have been explored as liquid biopsy biomarkers in DLBCL particularly targeting nucleic acids such as circulating RNA, noncoding RNA, and miRNA [31,32,33]. Concerning EV proteins, previous studies used predefined well-studied lymphoma-related biomarkers [34,35]. However, discovery-based quantitative global proteomic analysis of DLBCL-EVs is lacking. For this reason, we have analyzed by MS the proteome of EVs and whole-cells lysates, obtained from DLBCL cell lines. These small EV-containing fractions potentially contain vesicles originating from late endosomes (“exosomes”) and vesicles originating from the cell surface (plasma membrane), with both classes sharing common molecular players, including the ESCRT components TSG101, VPS4, and/or PDCD6IP (Alix) [36]. DLBCL-secreted EVs isolated in the current study presented an average size between 50 and 200 nm (Figure 1 and Table 1). The UC-suc isolation method produced EVs fractions with protein content enriched for membrane vesicle related compartments and trafficking such as phagocytic vesicle membrane, lysosomal membrane, recycling endosome membrane, and late endosome (Figure 3). We have previously developed an MS-based tool to assess quality control (QC) of EV fractions [12]. Characterization of EVs in the current study (Figure 1c) displayed concordance with MISEV2018 guidelines when evaluating known markers of extracellular vesicles, such as tetraspanins CD9, CD63, and CD81 and proteins involved in EVs biogenesis [16]. This approach poses considerable advantages by simultaneously accessing general EV markers, subcellular markers, and contaminants. Alix protein, heat-shock proteins HSPA8 and HSP90AA1, and actin were among the most abundantly expressed proteins in DLBCL EVs (Figure 1). We have also identified TSG101, HSPA6, SCDBP, and CD81 at a medium to high protein expression level (0.5–0.75). Considering tetraspanin proteins, CD9 and CD63 together with CD81 expression levels showed considerable variability in the different DLBCL EV fractions, suggesting caution concerning evaluation of EV purity based solely on this subset of protein markers (Figure 1c). On the other hand, Golgi, mitochondrial and nuclear compartment markers represented by GOLGA2, BCL2, and NUP98, respectively, ranked 0 expression level, suggesting a low level of contamination from other subcellular compartments in the isolated EVs (Figure 1c). Altogether, our quality control suggests that the isolated DLBCL EVs-containing fractions are bona fide for further analysis and extrapolation for selection of biomarkers of the ABC subtype, which are applied as surrogate markers for poor prognosis.

The total numbers of identified proteins were significantly different between cell lines and EVs (Figure 2). RIVA and OCI-ly3, which are classified as ABC subtype, displayed large differences in the total number of proteins identified in the whole-cell proteome. OCI-ly3 has the highest proliferative rate which may explain the difference in the exceedingly high number of proteins identified in OCI-ly3. In fact, among the unique proteins for OCI-ly3, the most representative biological process was mRNA transcription. This protein heterogeneity suggests that proteome analysis of DLBCL tumor tissue may also contribute to further subclassify DLBCL subtypes. It is known that EVs constitute subpopulations with varying protein cargo, which explains lower protein identifications in EVs compared to cells. In this view, we can speculate that in a baseline status the content of EVs is rather stable and the variation on EV cargo is likely dependent on stimuli. Furthermore, EVs contain a few unique proteins that are not identified in whole cells. The classification of EV populations is difficult due to lack of specific markers for assignment of the different components that copurify. Among the species that are copurified along with EVs are viral particles, protein aggregates, and neutrophil extracellular traps (NETs). The presence of nucleoplasm, which mainly comprises histone proteins, in EVs may be associated with a high level of DNA, as previously reported [37,38]. It remains to be resolved if these histone proteins are an integral part of EVs or copurified neutrophil extracellular traps (NETs)-like particles.

We have compared the global proteome of DLBCL whole-cell and secreted EVs between ABC and GCB molecular subtypes (Figure 4). We observed the B-cell receptor (BCR) signaling pathway, Fc gamma R-mediated phagocytosis, endocytosis, ErbB signaling, and natural killer cell-mediated cytotoxicity pathways were more enriched for regulated proteins in DLBCL EVs compared to the whole-cell proteome (Figure 5). We speculate that the enrichment of natural killer cell-mediated cytotoxicity pathway proteins is a result of protein overlap with the other main pathways (Figure A4). This indicates that immune response components are privileged for cargo into EVs (Figure A4). The ABC subtype of DLBCL relies on BCR signaling, and this pathway was observed more abundantly in ABC EVs than whole ABC cells (Figure A9 and Figure A10). Bruton’s tyrosine kinase (BTK), a BCR signaling component, is essential for the survival of ABC DLBCLs both in the absence and presence of mutations in the BCR activation pathway (e.g., mutation of CARD11 and CD79B proteins) [39,40]. BTK was significantly upregulated in ABC EVs (Figure A10), suggesting a potential role of EVs in disease dissemination as previously described for MyD88 L265P, which was demonstrated to support the survival of lymphoma cells by activation of BTK [31,41,42]. Further analysis of BCR-containing differential expressed proteins in EVs and whole cells (Figure A3) revealed that 18 out of 21 of those proteins were upregulated in ABC EVs. On the other hand, the numbers of upregulated and downregulated proteins for ABC whole-cell proteome were similar. BCR pathway inhibitors, including the BTK inhibitor ibrutinib, induce remissions in a subset of ABC DLBCL patients [43]. Moreover, upregulation of BCR pathway components in ABC EVs suggests it to be a potential target for therapy since the loss of B-cell receptor (BCR)- and phosphatidylinositol 3-kinase (PI3K)-activating proteins enhanced sensitivity. By contrast, the loss of negative regulators of these pathways (e.g., TRAF2, TNFAIP3) promoted resistance to MALT1 inhibitors [44]. EVs-containing BCR components are of extreme relevance since immunotherapy options targeting BCR components and auxiliary molecules such as the case of newly CD22 CAR T-cell therapy [45] can function as decoys, leading to disease refractoriness [46]. This was reported for CD20 (MS4A1 gene) upon the treatment with monoclonal antibody Rituximab using in vitro cultured cell lines and extracellular vesicles from patient’s blood [45]. We confirmed the presence of CD20 in DLBCL EVs by MS, except in HT EVs which lack its expression also at the cellular level (Figure A8). Overall, our data suggest that EVs from DLBCL can be exploited for patient sub classification, patient follow-up, and minimal residual disease monitoring.

Previous analysis of whole-cell global proteome of DLBCL, comparing the ABC and GCB molecular subtypes, demonstrated that the two subtypes could be distinguished by a 55 protein signature [6]. In our study, we have identified a considerably higher number of DEPs (288 at adj. *p* value ≤ 0.05) between the ABC and GCB cellular proteome. This discrepancy is likely attributable to the MS methodology used in the different studies and the number of cell lines and replicas. A detailed comparison of significantly regulated proteins between the two studies identified the regulation of similar proteins (Figure A5a,b). The overlap between significantly regulated proteins from our study and transcriptomics biomarkers in NanoString was three for cells and one for EVs (Figure A5c,d). Considering that the NanoString technology only targets 20 transcripts and that the input material used in NanoString is tumor tissue, including diverse cells from the microenvironment, the overlap between the results is reasonable. DLBCL EVs DEP in each pathway showed overlapping proteins between the five most enriched pathways such as proto-oncogene tyrosine-protein kinase SRC, E3 ubiquitin-protein ligase CBL, Casitas B-lineage lymphoma proto-oncogene b CBLB, and serine/threonine protein kinase PK3CA (Figure A4). CBL small family of Cbl ubiquitin E3 ligases, c-Cbl, Cbl-b, and Cbl-c regulates signaling through its N-terminal tyrosine-kinase-binding (TKB) domain [47]. Cbl proteins interact with tyrosine kinases through its TKB domain such as v-src oncogene, a preferential target of Cbl-c for degradation [48]. Once again, the oncogenic loading of EVs-ABC suggests that EVs have a potential use for disease monitoring. Follow-up experiments to evaluate the role of the observed oncoproteins and tumor suppressors listed in Figure A6 could elucidate if the function of these proteins are transferable between tumor cells.

Complete functional regulation analysis tests proteins matching a specific functional category both for significant enrichment and regulation [49]. Applying complete function regulation analysis on the obtained quantitative data from MS revealed that the Gleevec pathway is both significantly regulated between ABC and GCB and significantly enriched. Further scrutiny revealed that this regulation was driven by upregulation of Gleevec pathway proteins in the OCI-ly3 cell line and therefore not specific for both ABC cell lines.

In addition, we found CD44 among the top regulated proteins in EVs. CD44 expression in tissue was proposed as predictive of two-year mortality in patients with HIV-related DLBCL [50]. HIV-related DLBCL is comparatively more aggressive than DLBCL in HIV uninfected patients, which suggests that CD44 may also be associated with poor outcomes in the ABC subtype [51].

In total, 33 DUBs were identified, of which three were significantly regulated (Figure 6). Seven of the thirty-three DUBs are druggable with compounds developed by several pharmaceutical companies, in which several clinical trials are undergoing [22]. USP14 appears as an interesting general target in DLBCL, since it was found to be highly expressed in all cell lines (Figure 6). Our in vitro model findings correlate with studies on formalin fixed paraffin embedded tissue of DLBCL patients in which both expression of USP14 and UCHL5 was observed in more than 74% of the cases [52]. Previously, it has been demonstrated that b-AP15 inhibits the activity of USP14 and UCHL5, leading to ABC- and GCB-DLBCL cell apoptosis and suppression of cell migration [53]. On the other hand, UCHL1 was significantly upregulated in GCB subtype (log_2_ FC 26.66; Adj. *p*-value 1.75 × 10^−24^). In general, UCHL1 is induced in B cells undergoing the GC reaction and is also highly expressed in lymphomas derived from GCB cells, providing an essential survival signal. Particularly, UCHL1 was associated with a more aggressive form of the DLBCL-GCB type [54].

In addition, E3 ligases were highly expressed in all conditions and might also serve as promising targets for DLBCL (Figure A7). Distinct E3 ligases were specifically upregulated in ABC such as UHRF1, NEDDL4, TCEB3, and RING1. On the other hand, TRIM22, ARIH1, and BCL6 interactor of the Cullin-3 (Cul3), a major component of a multimeric E3 ubiquitin ligase complex were mainly abundant in HT. Among the NanoString gene expression, we found proteostasis factors, namely SERPINA9 (the serine protease inhibitor, Germinal center B-cell-expressed transcript 1 protein), UBX4 (UBX domain-containing protein 4, part of the ubiquitin-dependent ERAD pathway), and TRIM56 (E3 ubiquitin-protein ligase important for protein K63-linked ubiquitination).

Applying a stringent cutoff using adjusted *p* values below 0.05 resulted in 15 proteins differentially regulated between ABC and GCB that were common for cells and EVs. These 15 proteins were regulated in the same direction for both cells and EVs in 13 out of 15 cases. Correspondingly, applying a less stringent cutoff (regular *p* value below 0.05) resulted in 105 proteins commonly regulated for cells and EVs. Again, 85 out of 105 proteins displayed the same direction of regulation for cells and EVs. We have determined that the protein content in DLBCL EVs reflects their cell of origin and its biology. For example, we demonstrated similar regulation of B-cell receptor signaling pathway in both cells and EVs. These findings lead us to conclude that the observed regulation between ABC and GCB in EVs partly reflects the regulation observed in cell of origin. Furthermore, we demonstrated protein regulation overlap with previous MS studies [6] based on a larger number of ABC and GCB cell lines, which demonstrated that ABC and GCB cell lines are stratifiable based on MS data (Figure A5). Overall, our results suggest EVs may play a broad role in pathobiology of DLBCLs and carry potential biomarkers for subtyping. We further speculate that EVs would prove useful in residual disease monitoring.

## 4. Materials and Methods

### 4.1. Cell Culture Sample Preparation

DLBCL cell lines of the ABC subtype (RIVA, and OCI-ly3) and GCB subtype (DB and HT) were obtained from Leibniz Institute DSMZ (German Collection of Microorganisms and Cell Cultures, Braunschweig, Germany) and were cultured following DSMZ recommendations. Cell culture was tested for mycoplasma contamination. In brief, cells were grown in RPMI medium (Invitrogen, Waltham, MA, USA) supplemented with 10% or 20% fetal bovine serum accordingly with supplier instructions. Cell lysis was performed using a RIPA buffer containing 2% SDS. The lysates were sonicated using a Branson type sonicator and cleared by centrifugation at 14,000× *g* for 20 min at 4 °C.

### 4.2. EV Isolation

EVs were isolated from 48 h growth conditioned media (200–450 million cells) supplemented with 10% or 20% ultracentrifugation EV-depleted FBS (UC-dFBS) as follows. First, centrifugation at 3000× *g* for 20 min at 4 °C, was performed followed by 12,000× *g* centrifugation, for 60 min at 4 °C, to remove unwanted dead cells and cellular debris. Clarified conditioned media was ultracentrifuged in a Beckman Coulter Optima (L-90K) at 100,000× *g* for 120 min at 4 °C with a Type 32.1 Ti rotor (k-factor: 229) to pellet EVs. The supernatant was carefully removed, and crude EV-containing pellets were resuspended in ice-cold phosphate-buffered saline (PBS), followed by floatation on sucrose cushion (30%, D2O) for 60 min at 100,000× *g* at 4 °C to remove non-EV protein complexes. After washing by pelleting the EVs collected in the sucrose cushion for 16 h at 100,000× *g* at 10 °C, EVs were resuspended in PBS, subjected to the BCA (colorimetric) assay, and stored at −80 °C until further use.

### 4.3. EVs Analysis by Transmission Electron-Microscopy

Purified EVs were absorbed onto formvar/carbon-coated glow discharged copper EM grids (5 μL on each grid) for 20 min and fixed with 2% formaldehyde (Hatfield, MA, USA) for 20 min. After washing in 10 drops of distilled water, the grids were stained with 2% Uranyl acetate (Hatfield, MA, USA) for 5 min. Transmission electron microscopy was performed using a FEI Tecnai G2 Spirit BioTWIN TEM operating at an accelerating voltage of 120 keV. Images were acquired using an Olympus-SIS Veleta CCD Camera.

### 4.4. EVs Characterization

Size and EVs particles’ concentration were analyzed by NS300 nanoparticle tracking analysis (NTA) system. Samples were prediluted to match the instruments measurements range, and both measures were normalized by the volume of sample.

### 4.5. Peptide Sample Preparation

Samples containing a minimum of 20 µg of total proteins of whole-cell and EVs lysates were further processed by the filter-aided sample preparation (FASP) method. In short, protein solutions containing SDS and DTT were loaded onto filtering columns (Millipore, Billerica, MA, USA) and washed exhaustively with 8M urea in HEPES buffer [55]. Proteins were reduced with DTT and alkylated with IAA. Protein digestion was performed by overnight digestion with trypsin sequencing grade (Promega, Madison, WI, USA).

### 4.6. Mass Spectrometry Analysis

Peptides samples were analyzed by nano-LC-MSMS (Dionex RSLCnano 3000) coupled to an Exploris 480 Orbitrap mass spectrometer (Thermo Scientific, Hemel Hempstead, UK). In brief, the samples (5 μL) were loaded onto a custom made fused capillary precolumn (2 cm length, 360 μm OD, 75 μm ID) packed with ReproSil Pur C18 5.0 µm resin (Dr. Maish, Ammerbuch-Entringen, Germany) with a flow of 5 μL per minute for 6 minutes. Trapped peptides were separated on a custom made fused capillary column (25 cm length, 360 μm outer diameter, 75 μm inner diameter) packed with ReproSil Pur C18 1.9-μm resin (Dr. Maish, Ammerbuch-Entringen, Germany) with a flow of 250 nL per minute using a linear gradient from 89% A (0.1% formic acid) to 32% B (0.1% formic acid in 80% acetonitrile) over 56 min.

Mass spectra were acquired in positive ion mode applying automatic data-dependent switch between one Orbitrap survey MS scan in the mass range of 350–1200 m/z followed by higher-energy collision dissociation (HCD) fragmentation and Orbitrap detection of fragment ions with a cycle time of 2 s between each master scan. MS and MSMS maximum injection time were set to “Auto”, and HCD fragmentation in the ion routing multipole was performed at normalized collision energy of 30%, and ion selection threshold was set to 10,000 counts. Selected sequenced ions were dynamically excluded for 30 s.

### 4.7. Protein Identification

The obtained MS data from the 48 LC-MS runs were searched using VEMS [56]. Database dependent search against a standard human proteome database from UniProt (3AUP000005640) was performed including permutated protein sequences, where Arg and Lys were not permutated. A maximum of 4 missed trypsin cleavages were used. Carbamidomethyl cysteine was included as fixed modification. Methionine oxidation, N-terminal protein acetylation, lysine acetylation, serine, threonine and tyrosine phosphorylation were included as variable modifications. Here, 5 ppm mass accuracy was specified for precursor ions and 0.01 Dalton for fragment ions. The false discovery rate (FDR) for protein identification was set to 1% for peptide and protein identifications. No restriction was applied for minimal peptide length for VEMS search. Identified proteins were divided into evidence groups as defined by Matthiesen et al. [57]. In addition, database dependent search was performed using all human, bacteria, and virus proteins sequences in UniProt. The number of nonhuman peptides was in the per mille range, and none of these peptides matched species that are associated to cell culture contamination. The result from this search further validated that no microorganism contamination of the cell cultures was present (confirming routine mycoplasma examination).

### 4.8. Quantitative Analysis

Intensity-based absolute quantification (iBAQ) was calculated by dividing total ion counts with the number of theoretical canonical tryptic peptides for a given protein (missed cleavage theoretical peptides were not counted) [58]. Quantitative data from VEMS were analyzed in R statistical programming language. Protein label-free expression values were preprocessed by removing common MS contaminants followed by log_2_(x + 1) transformation, and quantile normalization. The quantitative values were all subjected to statistical analysis utilizing the R package limma [59] where contrast between ABC and GCB was specified (Appendix A whole cell and Appendix A for EVs). Correction for multiple testing was applied using the method of Benjamini and Hochberg [60]. The result from PCA analysis using R base function was performed by inputting all quantitative values without any biased pre-filtering. Volcano plots were plotted using the R package “EnhancedVolcano” (https://github.com/kevinblighe/EnhancedVolcano, accessed on 15 July 2021).

### 4.9. Functional Analysis

Functional enrichment based on the hypergeometric probability test in R has previously been described [49,61]. In all cases, the functional categories cellular component (CC), biological process (BP), molecular function (MF), and KEGG were analyzed based on all significant regulated proteins (*p* value < 0.05). In addition, functional enrichment analysis was performed based on proteins up- and down-regulated (*p* value < 0.05 and at least two-fold regulation). Appendix A summarizes the results from the functional enrichment analysis.

### 4.10. Protein Based Quality Control Plot for EVs

Average iBAQ values were averaged for each of the EVs sample groups DB, RIVA, OCI-ly3, and HT. Next, the average iBAQ values were ranked from 0 to 1. Next, the ranked expression of frequent exosome markers reported in the literature, and the 10 most frequently reported proteins in ExoCarta were extracted for the heatmap. The reference data for EVs from cell lines were obtained by two large studies using different EV enrichment methodologies: (1) PEG-based precipitation [18] and (2) differential ultracentrifugation [19].

## 5. Conclusions

We demonstrated that there is a potential for EVs proteome of DLBCL to provide signatures for DLBCL patients molecular subtyping. This opens avenues for exploring EVs proteome in liquid biopsies for signatures with diagnostic and prognostic value.

## Figures and Tables

**Figure 1 ijms-22-11004-f001:**
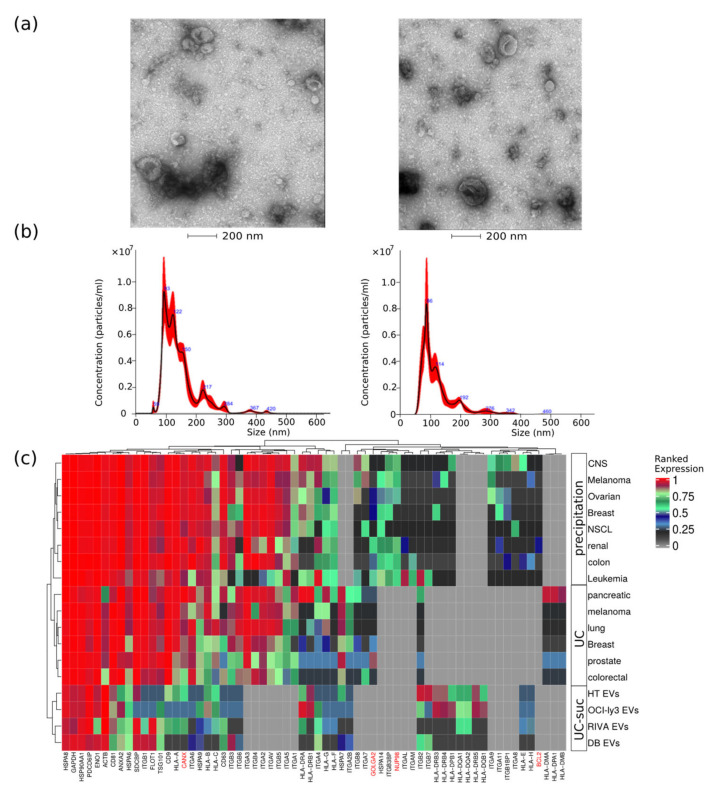
EVs quality control assessment evaluated by different methods. (**a**) Negative staining transmission electron microscopy (TEM); (**b**) nanoparticle tracking analysis of representative EV samples; and (**c**) analysis of EVs protein expression values obtained from different cancer cell types using three isolation methods, namely, precipitation, ultracentrifugation (UC), or UC combined with sucrose cushion (UC-suc). Cell organelle markers are indicated in red.

**Figure 2 ijms-22-11004-f002:**
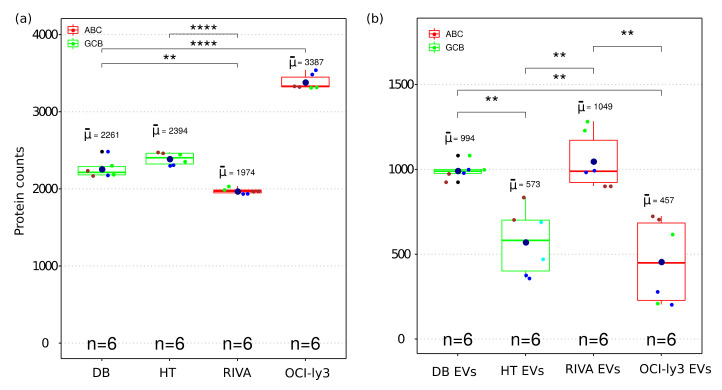
Boxplot of the number of identified proteins in whole cell (**a**) and EV (**b**) proteome from four DLBCL cell lines representing GCB and ABC sub types. The boxes indicate the median, second and third quartile. Mean values are represented by the large dark blue circles, while data points representing individual LC-MSMS runsare represented by smaller circles. All biological replica are depicted by a different small circle color and technical replica by identical small circle color. (**) correspond to *p*-value <0.01. (****) correspond to *p*-value < 0.0001.

**Figure 3 ijms-22-11004-f003:**
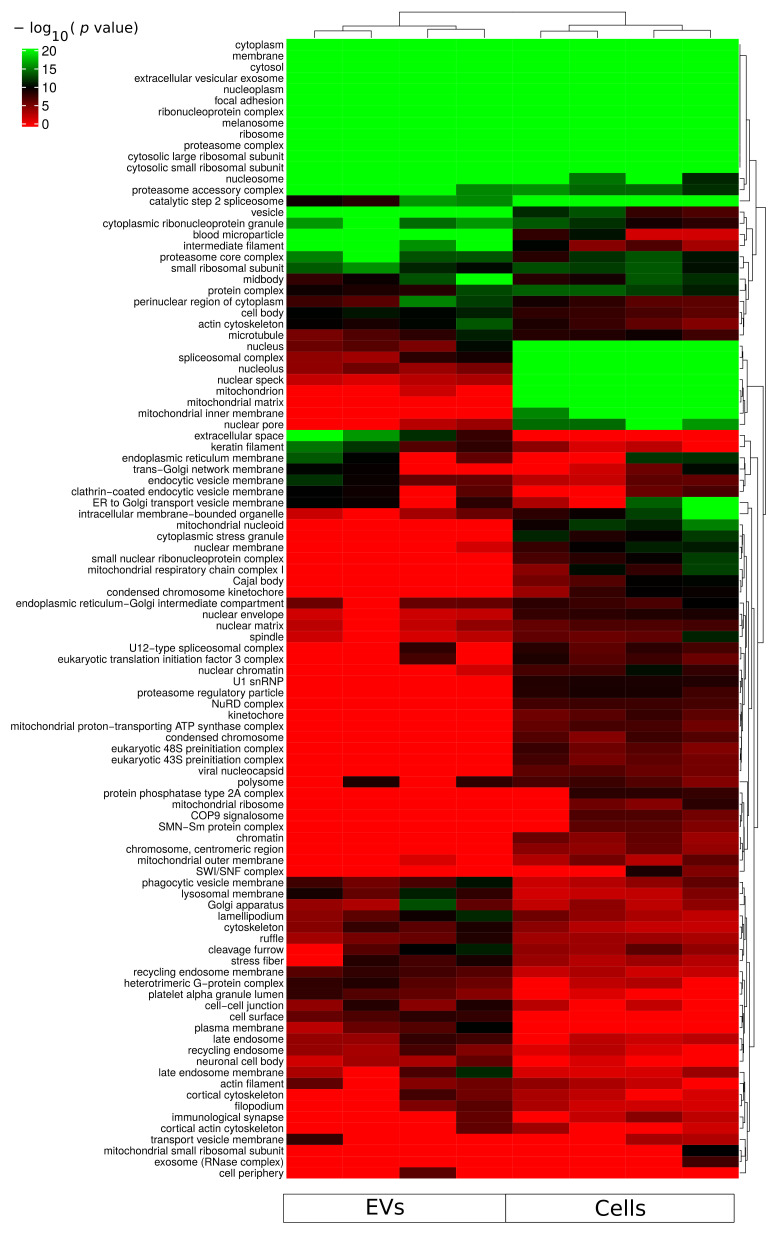
Functional enrichment for cellular component using all identified proteins from each sample group (cells and sEVs). A hypergeometric *p*-value threshold of 10^−6^ of at least one sample group was applied for construction of the heat map.

**Figure 4 ijms-22-11004-f004:**
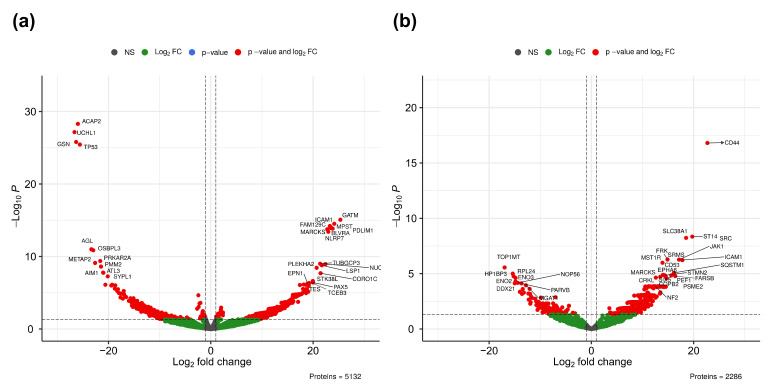
Volcano plots based on quantified proteins in whole proteome (**a**) and EVs (**b**). NS, nonsignificantly expressed proteins (grey); nonsignificantly differentially expressed proteins with at least two-fold regulation (green); and P & Log_2_ FC, significantly expressed proteins (red).

**Figure 5 ijms-22-11004-f005:**
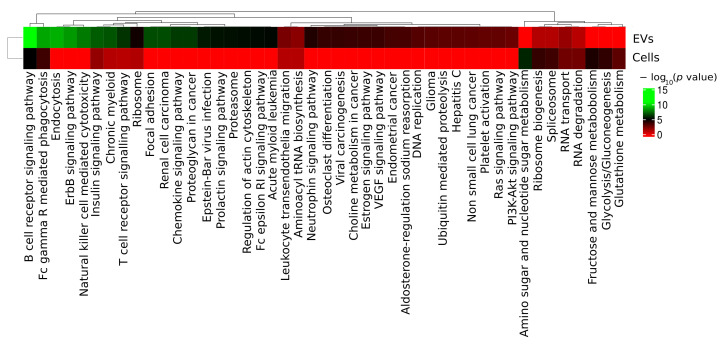
Overview of KEGG pathways enriched among significantly differential regulated proteins in DLBCL EV proteins and whole-cellular proteome. KEGG pathways colored in green reflect the high abundance of proteins in each category as opposing to pathways colored in red measured by −log_10_ (*p* value).

**Figure 6 ijms-22-11004-f006:**
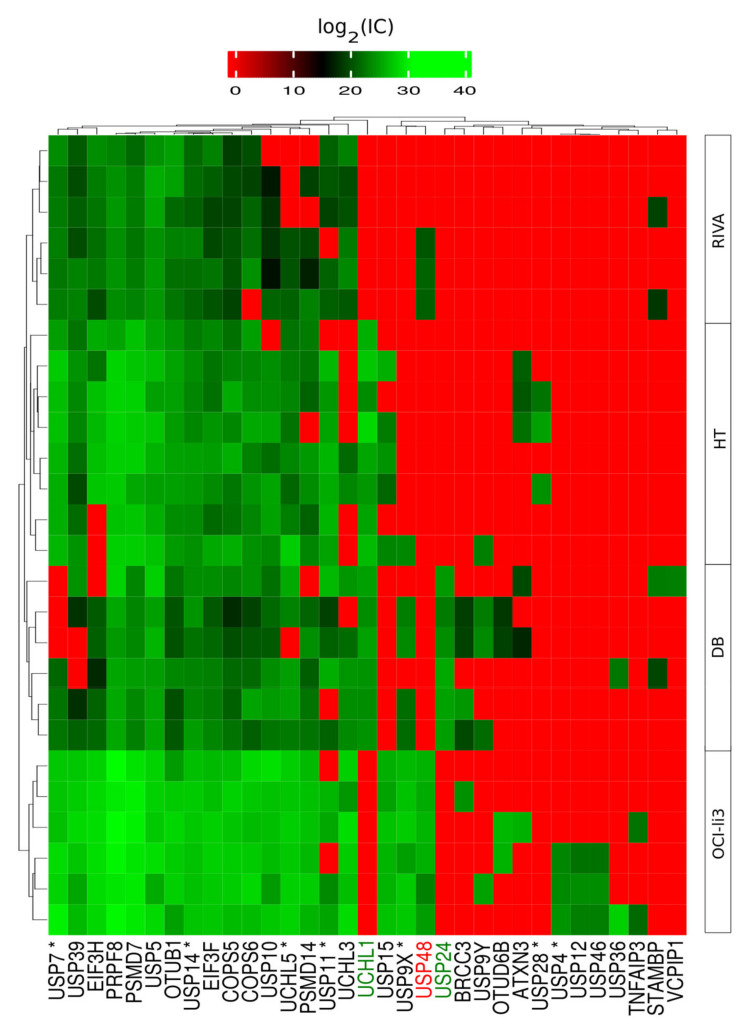
Heatmap of Log_2_ ion counts for DUB proteins identified in DLBCL whole cells. (*) indicates DUBs for which inhibitors have been developed are marked with an asterisk. Protein names in green correspond to DUBs upregulated in GCB and in red upregulated in ABC.

**Table 1 ijms-22-11004-t001:** EVs sample characteristics recorded by nanoparticle tracking analysis.

CELL LINE.	Mean Size (nm)	Mode (nm)	Particles/mL
DB	141.0 ± 2.0	105.7 ± 6.0	3.07 × 10^11^ ± 1.89 × 10^10^
HT	146.0 ± 7.1	98.7 ± 11.3	1.05 × 10^11^ ± 8.6 × 10^9^
RIVA (RI-1)	120.1 ± 6.6	81.5 ± 4.2	2.22 × 10^11^ ± 5.1 × 10^9^
OCI-ly3	126.9 ± 4.8	109.5 ± 9.5	1.46 × 10^11^ ± 1.46 × 10^10^

## Data Availability

The mass spectrometry proteomics data have been deposited to the ProteomeXchange Consortium via the PRIDE [62] partner repository with the dataset identifier PXD028267.

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
