# Peer review of "Proteomic Landscape of Extracellular Vesicles for Diffuse Large B-Cell Lymphoma Subtyping"

_ijms, 2021, doi:10.3390/ijms222011004_

Round 1

Reviewer 1 Report

In the manuscript, "Proteomic landscape of extracellular vesicles for diffuse large B-cell lymphoma subtyping" the authors define the whole cell and secreted extracellular vesicle (EV) proteome to distinguish the subtypes of DLBCL based on the cell of origin. Cell lines specific for DLBCL subtypes GCB and ABC were used in the study and the proteome was evaluated using nano-LC-MSMS and compared between GCB and ABC subtypes. Authors propose deubiquitinases as one of the potential drug targets to treat DLBCL. The study addresses an important area concerning the role of EVs, which can be isolated from liquid biopsies, in DLBCL pathologies.

However, a clear connection between EV proteome that distinguishes cell lines on the basis of cell of origin category is lacking and therefore needs to be emphasized.

Minor comments:

Lines 178-181: Please rewrite. Authors mean cellular compartment                                           associated proteins....

Line 200-201: proteomics data revealed significant differences in 288/1115...

Line 462: ..cleared by centrifugation

Manuscript sometimes is difficult to understand. Some examples are:

Lines 367-368; Lines 390-392; Lines 428-429 

Author Response

We thank reviewer one for the useful comments that enabled us to improve the manuscript. We have also amended the manuscript text according to the reviewer suggestions using track changes.

Concerning, the reviewer's comment on "a clear connection between EV proteome that distinguishes cell lines on the basis of cell of origin category is lacking and therefore needs to be emphasized".

This is a good point, and we agree that the discussion does not summaries well all the observations that led us to this conclusion. The below section was added to the discussion:

Applying a stringent cut off using adjusted P values below 0.05 resulted in 15 proteins differentially regulated between ABC and GCB that were common for cells and EVs. These 15 proteins were regulated in the same direction for both cells and EVs in 13 out of 15 cases. Correspondingly, applying a less stringent cut off (regular P value below 0.05) resulted in 105 proteins commonly regulated for cells and EVs. Again, 85 out of 105 proteins displayed same direction of regulation for cells and EVs. We have determined that the protein content in DLBCL EVs reflects their cell of origin and its biology. For example, we demonstrated similar regulation of B cell receptor signaling pathway in both cells and EVs. These findings lead us to conclude that the observed regulation between ABC and GCB in EVs partly reflects the regulation observed in cell of origin. Furthermore, we demonstrated protein regulation overlap with previous MS studies [6] based on a larger number of ABC and GCB cell lines which demonstrated that ABC and GCB cell lines are stratifiable based on MS data (Figure A5). Overall, our results suggest EVs may play a broad role in pathobiology of DLBCLs and carry potential biomarkers for subtyping. We further speculate that EVs will prove useful in residual disease monitoring.

Reviewer 2 Report

The paper written by Ana Sofia Carvalho et al. deals about the proteomic landscape of cell lysates and EVs from diffuse large B-cell lymphoma subtypes, namely GCB and ABC. Two different cell lines for each subtype were used in the study and the complement of the identified proteins functionally analyzed. The paper is quite interesting, and the methodology is robust.

Despite this, the results are too speculative and no validation in patients or different cell lines has been done. This makes the work only potentially useful. Can the authors perform some western blotting assays on other cell lines of the same subtype, in the absence of patient sampling? These would increase the significance of their results.

Furthermore, part of the results and discussion is based on the possibility to use specific drugs to inhibit certain proteins. Can the authors confirm their hypotheses with some experiments?

Author Response

We thank for the reviewer's interest and comments. The reviewer's comments are interesting. However, to acquire new cell lines and isolate EVs by differential centrifugation followed by sucrose cushion with three biological replicas for each cell line will likely take us minimum a year. We lose about 80% of our proteins in the sucrose cushion step, meaning that we start out with rather large cell cultures of 250-450 million cells. The drug experiments proposed by the reviewer are currently pursued in the laboratory and is planned for a new study that is going to run over the next one to two years. However, we can further mention the following study:

Recently, Rossi and co-workers (10.1038/s41375-021-01347-6), reported a cytotoxic effect of venetoclax, prexasertib or in combination on DLBCL cell lines, demonstrating that the results depicted in Figure A8, for the venetoclax are bona fid.

Round 2

Reviewer 2 Report

Authors did not experimentally consider any of the reviewer's suggestions. I believe that although the work is scientifically sound, it is less strong from an application point of view.